# Wi-Fi Fingerprint Indoor Localization by Semi-Supervised Generative Adversarial Network

**DOI:** 10.3390/s24175698

**Published:** 2024-09-01

**Authors:** Jaehyun Yoo

**Affiliations:** School of AI Convergence, Sungshin Women’s University, 34 da-gil 2, Bomun-ro, Seongbuk-gu, Seoul 02844, Republic of Korea; jhyoo@sungshin.ac.kr; Tel.: +82-2-920-7695

**Keywords:** generative adversarial network, indoor localization, semi-supervised learning, Wi-Fi fingerprint

## Abstract

Wi-Fi fingerprint indoor localization uses Wi-Fi signal strength measurements obtained from a number of access points. This method needs manual data collection across a positioning area and an annotation process to label locations to the measurement sets. To reduce the cost and effort, this paper proposes a Wi-Fi Semi-Supervised Generative Adversarial Network (SSGAN), which produces artificial but realistic trainable fingerprint data. The Wi-Fi SSGAN is based on a deep learning, which is extended from GAN in a semi-supervised learning manner. It is designed to create location-labeled Wi-Fi fingerprint data, which is different to unlabeled data generation by a normal GAN. Also, the proposed Wi-Fi SSGAN network includes a positioning model, so it does not need a external positioning method. When the Wi-Fi SSGAN is applied to a multi-story landmark localization, the experimental results demonstrate a 35% more accurate performance in comparison to a standard supervised deep neural network.

## 1. Introduction

Indoor localization has attracted increasing attention for location awareness where the Global Navigation Satellite System (GNSS) does not work in indoor buildings. Many different methods have been developed by using methods such as Pedestrian Dead Reckoning (PDR) [1], hardware-based localizations such as Angle of Arrival (AoA) [2] and Time Difference of Arrival (TDoA) [3], and distance estimations [4]. Additionally, Wi-Fi Received Signal Strength Indicator (RSSI) fingerprint localization has become popular due to its advantage of utilizing complementary Wi-Fi RSSIs obtained from a large number of existing Access Points (APs) built into the structure [5]. A Wi-Fi fingerprint is defined as a labeled data point that is a pair of RSSIs and their measuring location. To estimate a location, given a set of RSSIs, a machine learning approach aims to find a mapping, such that
$$\begin{array}{c} \mathrm{Positioning}\;\mathrm{function}:a\;\mathrm{set}\;\mathrm{of}\;\mathrm{RSSIs}\to a\;\mathrm{location}. \end{array}$$

A major issue with Wi-Fi RSSI fingerprint localization is that data collection is costly. It needs manual collection across all positioning areas, and an annotation process to label locations to RSSI sets. To reduce these effort, a Generative Adversarial Network (GAN) might be one of the promising solutions. The purpose of the GAN is to produce artificial data samples similar to real ones [6]. A typical GAN has two independent deep neural networks, i.e, a generator and a discriminator. An adversarial learning approach via a min–max game is used for training; the generator is learned to fool the discriminator by making realistic fake data, whereas the discriminator is learned to distinguish fake and real data.

In the scenario where GANs are assumed to be applied for fingerprint indoor localization, a generator might be modeled to produce artificial RSSI data, such that
$$\begin{array}{c} \mathrm{GAN}\;\mathrm{generator}:\mathrm{noise}\to a\;\mathrm{set}\;\mathrm{of}\;\mathrm{RSSIs}. \end{array}$$

It is expected that the GAN improves the localization accuracy by supplying a large amount of training data. However, most of the existing GAN-based fingerprint methods [7,8,9,10,11] can only produce *unlabeled* RSSI samples. By this restriction, their methods have reported using only a few visible Wi-Fi APs whose locations are known preliminarily. This limitation is different to a general fingerprint localization environment in which a large number of location-unknown APs are used.

The main contribution of this paper is to develop a Wi-Fi Semi-Supervised GAN (SSGAN) for fingerprint localization, which produces synthetic *labeled* RSSI data, such that
$$\begin{array}{c} \text{Wi-Fi}\;\mathrm{SSGAN}\;\mathrm{generator}:a\;\mathrm{location}\to a\;\mathrm{set}\;\mathrm{of}\;\mathrm{RSSIs}. \end{array}$$
One of main differences from a normal GAN is the input configuration. By inputting a specific query location, the SSGAN generates a corresponding labeled RSSI fingerprint, whereas a normal GAN produces only unlabeled RSSI values irrelevant to locations. Moreover, the proposed Wi-Fi SSGAN includes an additional classification network, which can be utilized as a positioning model, so it does not need to employ an extra positioning method. The produced labeled data can help improve learning performance of the positioning network model because the fingerprint localization mainly uses labeled data. While GAN is an unsupervised learning method, SSGAN can be seen as a type of semi-supervised learning whose main purpose is to learn a classifier, as illustrated in Figure 1.

Because raw RSSI fingerprints are sparse due to the AP range limit, feature extraction from RSSIs [12] is mandatory for fingerprint localization. This paper applies an Auto-Encoder (AE) to convert them to trainable data, where an AE is an unsupervised deep representational neural network to recover original input data [13]. A learned AE model extracts feature values of neural nodes from the middle layer. The resultant feature set has far lower dimensionality than the original data set. As a result, the high dimensional and sparse raw RSSI measurements are transformed to feature sets by the AE, and then the feature sets are used as input data for the Wi-Fi SSGAN to learn an indoor localization model.

For the experimental study, we collected Wi-Fi fingerprints through corridors at a five-story office building, in which 508 different AP devices are scanned across five floors. To evaluate the proposed Wi-Fi SSGAN algorithm, we define a multi-classification problem as landmark localization. From the experiments, Wi-Fi SSGAN achieved 35% better accuracy compared to a supervised deep neural network when a small amount of training data was used.

The rest of this paper is organized as follows. Section 2 overviews the related works. Section 3 describes the Wi-Fi fingerprint data preprocessing. Section 4 mainly introduces the Wi-Fi SSGAN algorithm. Section 5 and Section 6 report the experimental results and conclusion, respectively.

## 2. Related Works

### 2.1. Data Generation for Wi-Fi Fingerprint Localization

To improve the cost efficiency of fingerprint indoor localization, various approaches have been proposed. Unlabeled data that include only RSSI measurements without ground-truth locations can be simply collected by crowdsourcing [12,14]. To exploit the unlabeled data, a semi-supervised learning method has been applied for indoor localization [15,16]. It first decides pseudo-labels (e.g., locations) of the unlabeled data based on a graphical representation, and then it learns a localization model with a penalty balance for the pseudo-labeled data.

The GAN can produce qualified fake data samples if a generative model and discriminative model are successfully trained to outplay it [10,17]. GAN learns its model by support of the generated data, as well as real data. It is easy to confirm the trustworthiness of fake data by visually comparing them to real ones. An accurately taught generator can produce fake data infinitely.

Because the standard GAN method has a weakness of convergence problem [18], the Wasserstein GAN (WGAN) was developed, in which the objective loss function is defined as the Wasserstein distance. Later, the WGAN-gradient penalty (GP) enhanced learning performance more by adding a gradient penalty to the loss objective [19], and it has recently been used for various applications such as images, text and digital signals.

### 2.2. Semi-Supervised GAN (SSGAN)

Compared to the original GAN, which uses only unlabeled data, the SSGAN [20] might exhibit a better learning result by utilizing labeled data as well. In [11,21], location information is used as the generator’s input to produce fingerprint data samples to improve the positioning accuracy. In [22,23], the SSGAN methods are validated to produce accurate synthetic data for applications of cross-modal hashing and clinical decision, respectively.

A more advanced strategy to apply to SSGAN learning is to involve a part of prediction network into a unified model. By combining a prediction model into a discriminator, it does not need to employ an extra prediction model. In [24], the SSGAN was used to generate labeled electroencephalography (EEG) signal samples, and had a validated better performance than a standard GAN.

Figure 1 compares the GAN and SSGAN. In the SSGAN, the label prediction is computed by a classifier that shares its network model with a discriminator, while the original GAN concentrates on producing unlabeled data without a prediction model.

## 3. Wi-Fi Fingerprint Preprocessing

This section overviews Wi-Fi fingerprint data localization in Section 3.1 and feature extraction by the AE in Section 3.2. The feature data will be used as input of the main learning algorithm described in Section 4.

### 3.1. Wi-Fi RSSI Fingerprint for Landmark Localization

For landmark localization, the Wi-Fi RSSI fingerprint data are collected by placing a receiver, e.g., a smartphone, at different landmark locations. A receiver measures RSSIs obtained from near APs that are broadcasting their signals periodically. The Wi-Fi signal conveys unique information of a AP transmitter by means of Media Access Control (MAC) address. This enables the receiver to recognize which AP sent the Wi-Fi signal. A general fingerprint localization method utilizes all RSSIs of APs scanned across the positioning area.

Suppose that *N* number of Wi-Fi RSSI fingerprints at *N* locations are obtained from total *d* number of APs, given by
(1)$$\begin{array}{c} D={\left\{\left(r_{i},y_{i}\right)\right\}}_{i=1}^{N}, \end{array}$$
where $r_{i}\in R^{d}$ is am RSSI fingerprint set and $y_{i}\in R^{l}$ is a landmark index. The landmark label index $y_{i}$ is defined as a *l*-by-1 one-hot vector, where *l* is the number of landmarks that are predesignated by a developer. The RSSI set at the *i*-th landmark is given by
(2)$$\begin{array}{c} r_{i}={\left[rssi_{i}^{1},rssi_{i}^{2},\cdots ,rssi_{i}^{d}\right]}^{T}, \end{array}$$
where $rssi_{i}^{j}$ is a RSSI measurement obtained from the *j*-th AP. The dimensionality of an RSSI set $r_{i}$ equals the number of APs, *d*.

Given the training data *D*, the objective of machine learning in the training phase is to build a classifier: $R^{d}\to R^{l}$, which represents a relationship between a set of Wi-Fi RSSI measurements and a landmark. In the test phase, when a query $r^{*}$ is given, the positioning model infers on which landmark a receiver is located.

### 3.2. Feature Extraction by Auto-Encoder (AE)

The outstanding property of raw RSSI fingerprint data is its sparsity as shown in Figure 2a. By restriction of the Wi-Fi signal propagation range, there are many empty elements in a raw RSSI set, so that $r_{i}$ in (Equation 2) has many empty values. Typically, these empty components are filled with a possible minimum value, such as −100 dBm. Because this prompts inaccurate learning performance, feature extraction for the sparse RSSI data is mandatory. The objective of feature extraction is to obtain a model *H* that converts a raw *d*-dimensional Wi-Fi RSSI fingerprint set $r\in R^{d}$ into a *s*-dimensional feature set $x\in R^{s}$ (d≫s), where *s* is dimensionality of feature data.

AE is an unsupervised deep representational neural network. It trains a meaningful feature space among input data in a layer-wise manner by learning a neural network model to replicate the original input data to output. Hidden layers from the input layer to the feature layer are called encoders, and the rest of the layers for the restoration are called decoders, as described in Figure 2b.

Given the Wi-Fi RSSI data set ${\left\{r_{i}\right\}}_{i=1}^{N}$ from (Equation 1), the encoder converts raw data $r\in R^{d}$ to low-dimensional feature data $x\in R^{s}$. The decoder reconstructs the feature x back to the original data r by estimating its prediction $\hat{r}$. More detail of mathematical explanation of the AE can be found in [13].

After the AE model is learned, the encoder part is used as the feature extraction method to obtain the following feature database,
(3)$$\begin{array}{c} O={\left\{\left(x_{i},y_{i}\right)\right\}}_{i=1}^{N}. \end{array}$$
The newly made database (Equation 3) replaces the original dataset (Equation 1) to learn the proposed Wi-Fi SSGAN model in the next section.

## 4. Wi-Fi SSGAN

The Wi-Fi SSGAN consists of multiple neural networks, and they are learned by complementary optimizations. Variables $\left(x^{f},y^{f}\right)$ and $\left(x^{r},y^{r}\right)$ are fake and real data, respectively. Figure 3 shows network models of Wi-Fi SSGAN in which the generative model produces fake labeled RSSI feature data, and the combined model of discriminator and classification predicts a location and distinguishes fake data, simultaneously.

### 4.1. Generator

The SSGAN generator produces fake samples $x^{f}∼P_{f}$ with respect to a specific location query $y^{f}$ by learning the network with parameter set $\theta_{G}$. Using an artificial label $y^{f}$ as the generator’s input is one of the major differences between the SSGAN and a normal GAN that only allows random noise as input. A simulated RSSI set generated by the SSGAN generator mimics a real fingerprint sample in relation to an actual location. Consequently, the accurately created fingerprint data can effectively support the learning of a positioning model.

### 4.2. Discriminator

When real data $x^{r}∼P_{r}$ are initially given and fake data $x^{f}∼P_{f}$ are made from the generator, the discriminator aims to recognize whether the produced samples are fake or not. The learning process lets the generator and discriminator outplay each other. The learning process is repeated until the generator finally produces realistic samples, so that the discriminator with $\theta_{D}$ does not exactly distinguish if the generated samples are fake.

### 4.3. Classifier

Classifier involvement into the discriminator network is another major difference between the SSGAN and a normal GAN in which a prediction model does not exist. The classifier with $\theta_{C}$ and discriminator with $\theta_{D}$ share a network model, except for the output layer. It is noted that the fake data $x^{f}$ also have ground-truths $y^{f}$, so that the prediction errors of the fake data can be calculated. Therefore, training both the actual and the fake data can improve the learning performance of the classifier.

### 4.4. SSGAN Formulation

In WGAN-GP [19], the discriminator *D* and generator *G* play a min–max game based on the Wasserstein distance V(D,G), which is a distance between distributions *D* and *G*, such that
(4)$$\begin{array}{cc} \min_{G}\max_{D}V\left(D,G\right) & =-E_{x^{r}∼P_{r}}\left[D\left(x^{r}\right)\right]+E_{z∼P_{z}}\left[D\left(G\left(z\right)\right)\right], \end{array}$$
where $x^{r}∼P_{r}$ are real data, $z∼P_{z}$ are noise vectors, and $P_{z}$ is uniform distribution with bound [0, 1]. To achieve a solution to (Equation 4), separate optimizations are performed to derive each generator and discriminator. The discriminator is learned by minimizing the following loss:(5)$$\begin{array}{c} L_{D}\left(\theta_{D},\theta_{G}^{*}\right)=V\left(D_{\theta_{D}}\left(x^{r}\right),G_{\theta_{G}^{*}}\left(z\right)\right)+\rho E_{\bar{x}∼P_{\bar{x}}}\left[(∥\nabla_{\bar{x}}D\left(\bar{x}\right){{∥}_{2}\;-\;1)}^{2}\right]. \end{array}$$
In (Equation 5), the first term is the Wasserstein distance, and the second term is a gradient penalty controller to improve the stability of the learning convergence, where ρ is a tuning parameter and $\bar{x}$ are samples lying on a straight line between $P_{r}$ and $P_{f}$ [19]. On the other hand, the generator is learned to fool the discriminator by reducing the following loss:(6)$$\begin{array}{c} L_{G}\left(\theta_{G},\theta_{D}^{*}\right)=E_{x^{f}∼P_{f}}\left[D_{\theta_{D}^{*}}\left(x^{f}\right)\right]. \end{array}$$
In (Equation 5) and (Equation 6), $\theta_{G}^{*}$ and $\theta_{D}^{*}$ are the fixed weight parameters, respectively, so that each network focuses on learning own model.

The proposed Wi-Fi SSGAN is extended from the WGAN-GP in a semi-supervised learning manner. One of the main differences between SSGAN and WGAN-GP is the input configuration of the generator. SSGAN generator’s input $\bar{z}∼P_{\bar{z}}$ is defined as a concatenation of noise and label, given by
(7)$$\begin{array}{c} \bar{z}={\left[z,y_{f}\right]}^{T}. \end{array}$$
Accordingly, fake data $x^{f}$ are made by the generator given by
(8)$$\begin{array}{c} x^{f}=G_{\theta_{G}}\left(\bar{z}\right), \end{array}$$
and the Wasserstein distance is defined as
(9)$$\begin{array}{c} V\left(D,G\right)=-E_{x^{r}∼P_{r}}\left[D\left(x^{r}\right)\right]+E_{\bar{z}∼P_{\bar{z}}}\left[D\left(G\left(\bar{z}\right)\right)\right]. \end{array}$$

The second difference is the prediction model involvement into the discriminator network for the purpose of increasing the accuracy of label prediction. The combined discriminator and classifier (CDC) model minimizes the following loss:(10)$$\begin{array}{c} L_{CDC}(\theta_{D},\theta_{C},\theta_{G}^{*})=L_{D}\left(\theta_{D},\theta_{G}^{*}\right)+L_{C}\left(\theta_{C},\theta_{G}^{*}\right), \end{array}$$
where $L_{D}$ is defined in (Equation 5), and $L_{C}$ is as follows:(11)$$\begin{array}{c} L_{C}\left(\theta_{C},\theta_{G}^{*}\right)=E\left[\log P\left(y^{r}|x^{r}\right)\right]+E\left[\log P\left(y^{f}|G_{\theta_{G}^{*}}\left(\bar{z}\right)\right)\right]. \end{array}$$
Finally, the generator loss of SSGAN with the fixed $\theta_{D}^{*}$ and $\theta_{C}^{*}$ is given by
(12)$$\begin{array}{c} L_{G}\left(\theta_{G},\theta_{D}^{*},\theta_{C}^{*}\right)=E_{\bar{z}∼P_{\bar{z}}}\left[D_{\theta_{D}^{*},\theta_{C}^{*}}\left(G_{\theta_{G}}\left(\bar{z}\right)\right)\right]. \end{array}$$

The overall training algorithm of the proposed Wi-Fi SSGAN is summarized in Algorithm 1. After the training process, the AE model and the classifier model are obtained. For the test when a query Wi-Fi RSSI set is given, the AE model first converts it to a feature set. Then, the classifier performs the probabilistic inference to localization.
**Algorithm 1** Wi-Fi SSGAN**Input:** Training dataset ${\left\{r_{i},y_{i}\right\}}_{i=1}^{N}$ with RSSI set $r_{i}\subset R^{d}$ and one-hot landmark label $y_{i}\subset R^{l}$.**Output:** An auto-encoder $:R^{d}\to R^{s}$, and a classifier $:R^{s}\to R^{l}$.**Feature extraction:** 1:Given the training dataset, learn the auto-encoder model and build the new feature dataset $\{{\left(x_{i},y_{i}\right\}}_{i=1}^{N}$ with $x_{i}\subset R^{s}$.**SSGAN:** 2:Initialize a generator with $\theta_{G}$, a discriminator with $\theta_{D}$ and a classifier with $\theta_{C}$. 3:**repeat** 4:    **for** j=1,2,…,k **do** 5:        Sample a batch of real data ${\left\{x_{i}^{r},y_{i}^{r}\right\}}_{i=1}^{M}$ from the feature dataset. 6:        Produce a batch of fake data ${\left\{x_{i}^{f},y_{i}^{f}\right\}}_{i=1}^{M}$ by the generator in (Equation 8). 7:        Do optimization to reduce loss $L_{CDC}$ in (Equation 10) and update $\theta_{D},\theta_{C}$. 8:    **end for** 9:    Produce a batch of fake data ${\left\{x_{i}^{f},y_{i}^{f}\right\}}_{i=1}^{M}$ by the generator in (Equation 8). 10:    Do optimization to reduce loss $L_{G}$ in (Equation 6) and update $\theta_{G}$. 11:**until** end of learning 12:Obtain the classifier with $\theta_{C}^{*}$.

## 5. Experiments

### 5.1. Setup

We collect the Wi-Fi RSSI data through the corridors in a five-story office building, where each floor has different shape, as shown in Figure 4. In total, 508 different AP devices whose locations are unknown are scanned, so the dimensionality of the raw Wi-Fi RSSI set is 508. Seventeen different landmarks shown in Figure 4 are defined at particular landmarks, and the machine learning aims to solve a multi-classification problem (17 classifications in this paper). We collect the RSSI measurements within a radius of 5 m from each landmark center to hold the diversity of the data and to avoid overfitting of the learning. A total of 100 data points for each landmark are prepared for each training and test set. To assess the usefulness of the fake data for fingerprint localization, we divide the training data points into subsets ranging from 10% to 100%. We then compare the positioning and learning performances for each data ratio.

In Wi-Fi SSGAN, the generator network has three layers, and all hidden layers have 50 neural nodes. The generator input $\bar{z}$ in (Equation 7) is combination of 10-dimensional uniform noise and 17-dimensional landmark one-hot $y_{f}$. Because the dimensionality of the feature set produced by the AE is 20, the generator output $x^{f}$ is also 20-dimensional. The CDC network whose loss function is defined in (Equation 10) is defined to share only the input layer. The discriminator has one hidden layer with 200 nodes. The classifier has three hidden layers with 200 nodes. The CDC network results in an 18-dimensional probabilistic vector, whose first element indicates the prediction score and the rest is the landmark prediction. To learn the network, an Adam optimizer and Xavier initializer are used. The size of batch *M* in Steps 6, 7, and 10 in Algorithm 1, which indicates the each sample size of real and fake data to learn each generator and CDC, is set as M=50. Additionally, iteration number *k* at Step 5 in Algorithm 1, and parameter ρ in (Equation 5), are set as k=5 and ρ=0.01.

As the baseline to evaluate the proposed Wi-Fi SSGAN, a supervised deep neural network (DNN) having five layers and 200 nodes is used. Because the main contribution of the Wi-Fi SSGAN is to generate labeled fake data to improve the learning performance, the outstanding difference from the DNN is expected when a small amount of training data are available. An unsupervised normal GAN, such as WGAN-GP, is not suitable for comparison in the same setup because it produces only unlabeled data, making fingerprint localization unfeasible.

Through the same AE model, the same feature data as the input of the DNN and the Wi-Fi SSGAN are fairly applied. The AE is designed to have five layers, including input and output layers. The first and second hidden layers have 200 and 20 neural nodes, respectively, so that dimensionality of a feature set is 20. The rectified linear unit activation function and Adam optimizer are used to learn the AE network.

### 5.2. Landmark Localization by Wi-Fi SSGAN

For performance evaluation, we change the number of labeled training data ratio from 10% to 100% to learn both the Wi-Fi SSGAN and DNN models, and the portion of the training data are randomly picked out of entire training data. Because test is performed 10 times, the accuracy is represented by the mean and standard deviation. Figure 5 shows the test accuracy of the proposed Wi-Fi SSGAN classifier and the compared DNN, according to change in the amount of used training data. The Wi-Fi SSGAN entirely outperforms the supervised DNN, and a noticeable difference is found when a small amount of labeled data are used. For example, when only 10% samples are used, the Wi-Fi SSGAN achieves 35% higher accuracy. Moreover, the supervised DNN shows large variations in accuracy due to an insufficient amount of training data, whereas the Wi-Fi SSGAN deviations are consistent regardless of the data ratio. Additionally, across all data ratio cases, the proposed algorithm outperforms the DNN due to the support of accurately produced synthetic data.

Figure 6 shows the loss value graphs of Wi-Fi SSGAN during learning iterations when 10% samples are used. We choose the 10% ratio samples because it is common for GAN models to fail when using a small amount of actual data. Successful learning with the 10% ratio implies that the other cases will also be successful. In Figure 6, the losses of real and fake data in (Equation 11) are shown in the top figure, and the CDC loss in (Equation 10) and the generator loss in (Equation 6) are are in the bottom figure. As the fake data and the classifier are updated at each iteration, the loss decreases and eventually converges, indicating successful learning termination.

It is possible to visually confirm the accuracy of the generator by comparing the fake and real data. Figure 7 shows the produced RSSI feature data (red line) and the real data (blue-dash line) according to variations in the amount of used training samples.

We present the results for two groups: (i) when a small amount of actual data (10%) is used, and (ii) when a large amount of actual data (100%) is used. The first group results are shown in Figure 7a–c, and the second group results are shown in Figure 7d, Figure 7e and Figure 7f, respectively.

In this paper, the first case is emphasized to effectively determine if the generated fingerprint data are useful for positioning. From Figure 7a–c, we observe that the generated data are not only diverse, but also closely align with the actual data’s distribution. Diverse data are more beneficial than overfitted data for learning a fingerprint positioning model for two reasons. First, natural RSSI values at a location are inconsistent due to external factors, such as noise and multi-path problems. Second, machine learning algorithms require diverse data rather than overfitted data. In the second case, when sufficient data are used for learning, as shown in Figure 7d–f, the generator also produces high-quality synthetic data.

Another way to evaluate the usefulness of the generated data are by comparing the shape of the fake data to the actual data that have the same location labels. Figure 7a,d are the RSSI feature values whose labels are annotated from the second landmark, Figure 7b,e are from the fifth landmark, and Figure 7c,f are from the eleventh landmark. The results indicate that the fake data and the actual data with the same labels are visually similar, which helps maintain localization accuracy, even when only a small amount of actual data are used.

## 6. Conclusions

The Wi-Fi SSGAN, which is a new semi-supervised learning version of a generative adversarial network for Wi-Fi indoor localization, was presented. The proposed method aims to produce artificial fingerprint data to support a lack of actual fingerprint data. From the experiments, the similarity of the fake data to real data was demonstrated, and localization accuracy was improved, especially when small amounts of actual training data were used. Many RSSI-based indoor localization algorithms have been reported, and these methods typically depend on manually collected fingerprint-labeled data. Therefore, the accurately produced fingerprint data presented in this paper can significantly support RSSI-based positioning algorithms. For instance, the synthetic data can be used as an alternative for predicting locations in areas that are otherwise inaccessible. As a future work, we plan to further analyze the effectiveness of the generated data in enhancing other RSSI-based localization methods, and to explore its limitations in various scenarios where it may not be useful for positioning.

## Figures and Tables

**Figure 1 sensors-24-05698-f001:**
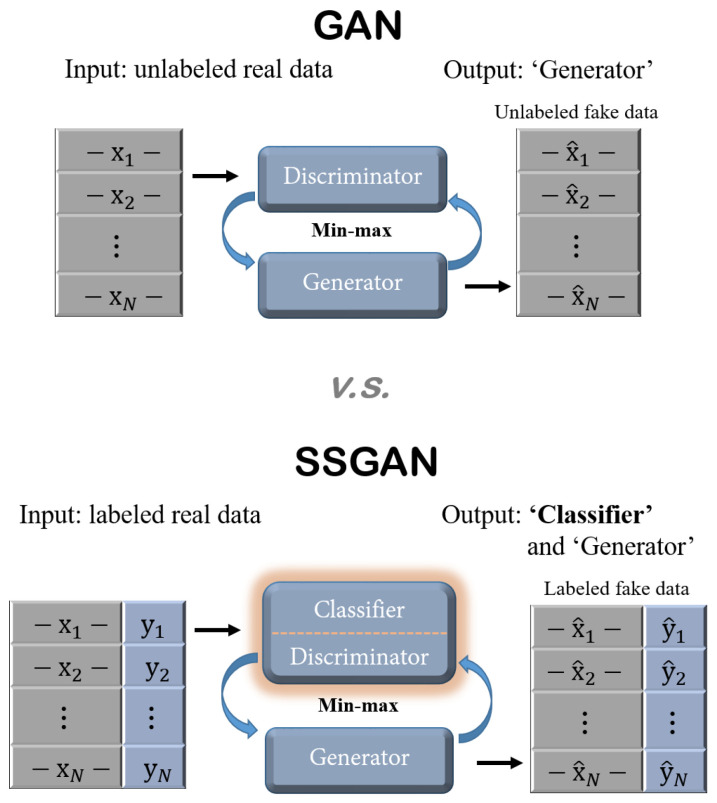
Comparison of GAN and SSGAN.

**Figure 2 sensors-24-05698-f002:**
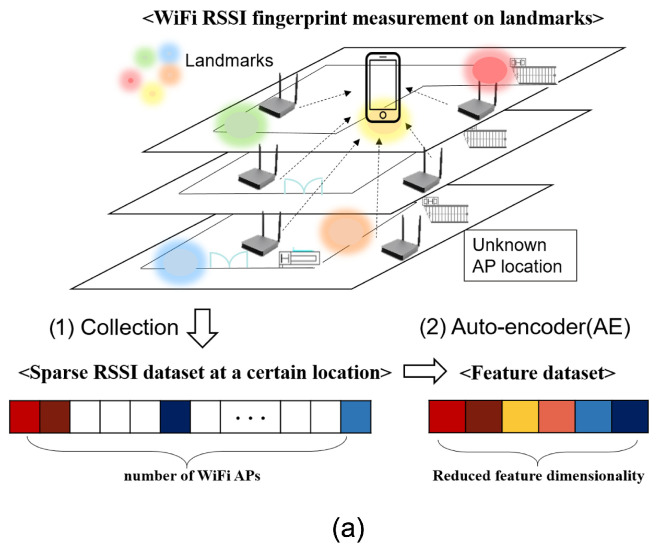
Wi-Fi RSSI fingerprint collection and feature extraction via auto-encoder (AE) in (**a**) and neural network structure of AE in (**b**).

**Figure 3 sensors-24-05698-f003:**
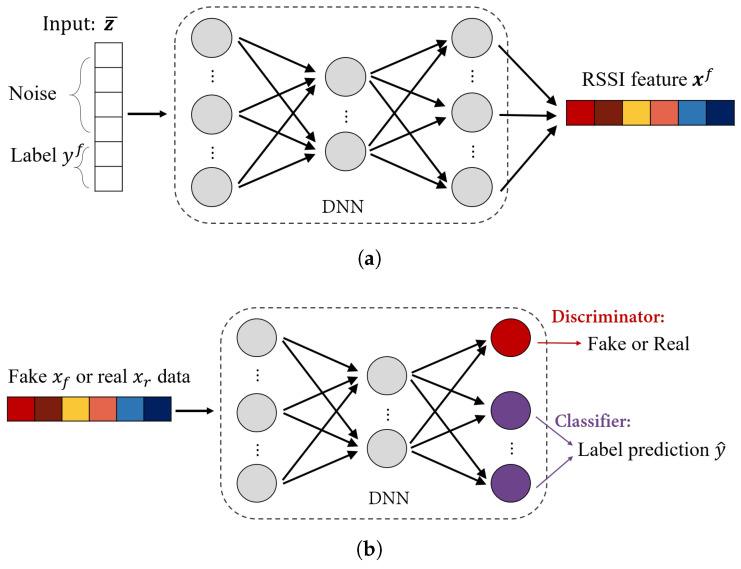
Wi-Fi SSGAN models: the generator (**a**), and the combined discriminator and classifier (**b**).

**Figure 4 sensors-24-05698-f004:**
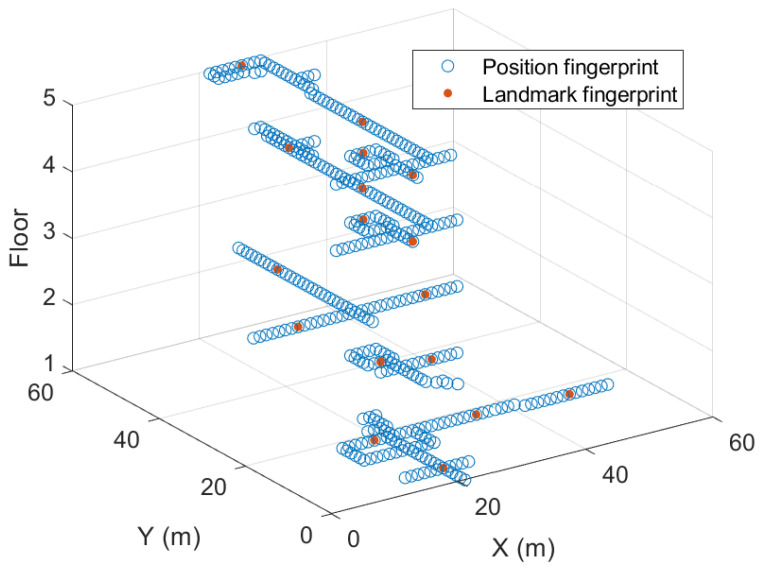
Fingerprint distribution from the experimental office building.

**Figure 5 sensors-24-05698-f005:**
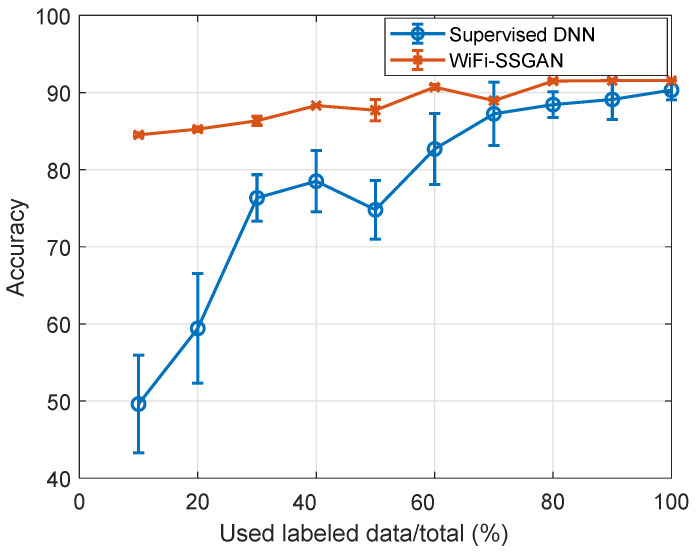
Comparison of classification performance according to changes in the amount of labeled data.

**Figure 6 sensors-24-05698-f006:**
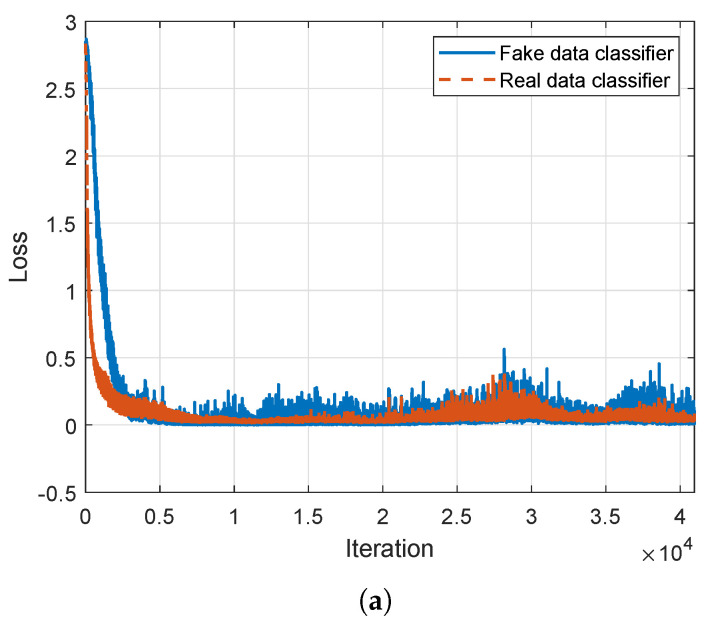
Learning curve of the proposed Wi-Fi SSGAN; classifier loss (**a**), and CDC and generator losses (**b**).

**Figure 7 sensors-24-05698-f007:**
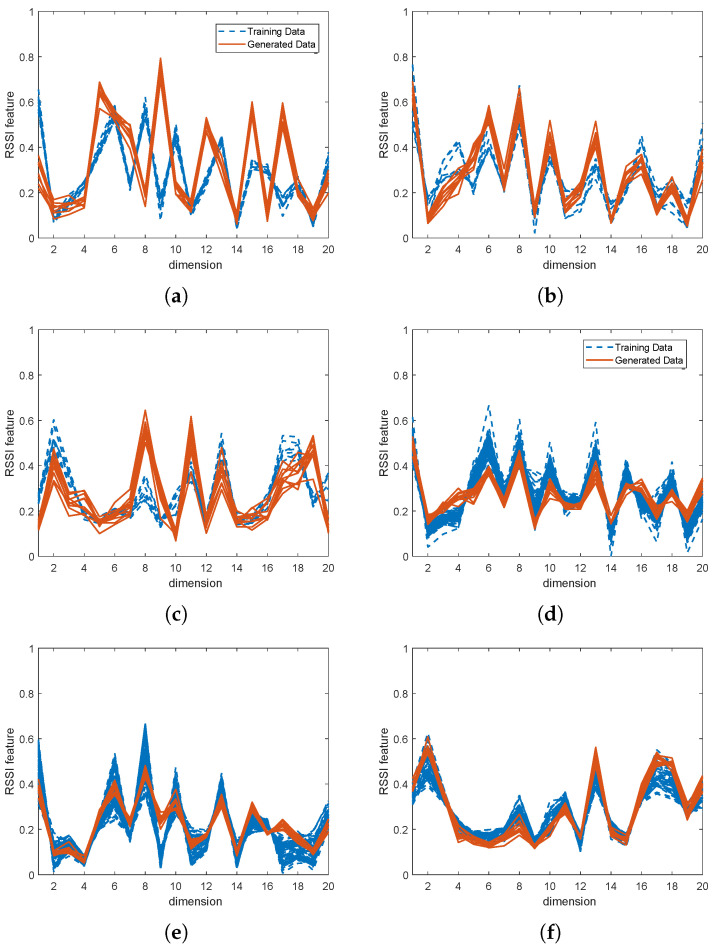
Synthetic labeled data by Wi-Fi SSGAN vs. actual labeled data; (**a**) landmark index = 2, sample ratio = 10%, (**b**) landmark index = 5, sample ratio = 10%, (**c**) landmark index = 11, sample ratio = 10%, (**d**) landmark index = 2, sample ratio = 100%, (**e**) landmark index = 5, sample ratio = 100%, (**f**) landmark index = 11, sample ratio = 100%.

## Data Availability

Data are contained within the article.

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
