# Peer review of "Wi-Fi Fingerprint Indoor Localization by Semi-Supervised Generative Adversarial Network"

_sensors, 2024, doi:10.3390/s24175698_

Round 1

Reviewer 1 Report

Comments and Suggestions for Authors

Dear Author

I have reviewed your paper with interest.

I think your paper is very well written.
However, there are some slight errors. Please correct the following two points.

1) Figure 2 contains two figures, but the indexes (a) and (b) are missing. Please include these.

2) Line 176 says "Figure ??" and does not include the figure number. Please include the figure number.

Sincerely yours

Author Response

Please find the attached pdf of the revision list.

Reviewer 2 Report

Comments and Suggestions for Authors

In this manuscript, the authors propose a deep neural network (GAN)-based approach to indoor localization relying on Wi-Fi fingerprinting data. The manuscript tackles an interesting topic and the overall quality of the research is good. However, I cannot recommend the paper for publication in present form, since there are some issues to clarify.

l. 1: please write correctly Wi-Fi (not WiFi).

l.16 and beyond: Put acronyms in capitals (Global Navigation Satellite System).

l.18: I suggest to also include, along with distance-based position algorithms, a little reference to angular based localization algorithms. For instance you’re discussing a GAN-based approach, which surely is not operating in close real-time. Therefore, I suggest you to consider referencing a work which considers a full-hardware AoA localization system operating in real-time (e.g. reconfigurable digital architectures for AoA estimation).

l.20: I don't think the claim of “not requiring additional devices” is well posed. In order to perform localization based on triliteration, it is necessary to have at minimum 3 devices to correctly estimate the x y z position of the target. Please clarify this.

l.35: The mathematical notation employed to define both the positioning function and the GAN generator are not really consistent. Please be rigorous or just use sentences (words).

l.54: I suggest to better highlight the contribution of this paper to the state-of-the-art in RSSI-based positioning. At the moment, it is not very clear. I hope you could make it more explicit.

l.248: The discussion of the results is too poor. Please elaborate a little bit more on what you have demonstrated.

Comments on the Quality of English Language

Minor english editing required.

Author Response

(The authors gave the same response as above.)

Reviewer 3 Report

Comments and Suggestions for Authors

Following are my evaluation for the manuscript, 'WiFi Fingerprint Indoor Localization by Semi-Supervised Generative Adversarial Network'
1. In the manuscript, the basic difference between the GAN and SSGAN remains unclear in the introduction; Furthermore, it would be helpful if authors can explain what difference does a labeled data adds against unlabeled RSSI data training.

2. Line 85 introduction, 'WiFi-SSGAN algorithm is extended from WGAN-GP to a semi-supervised learning version' did this work illustrate all three types of model? It is unclear. 

3. Section3.2, the explanation of feature extraction for SSGAN is incomplete and hard to understand. Even the variables used in equation are not defined. Same goes for equation 4.

4. Section 4, it would be helpful if explanation is more focused on 'how' the presented model is functioning to produce desired results.

5. If the training data is collected 17*100=1700 for 5 floors of the experimental setup; it implies the presented method is still ineffective and does not decrease manual labor and cost as it is claimed in abstract and introduction of the manuscript. 

6. The formulation of the model is done using simple GAN, WGAN-GP and proposed SSGAN however, the result section present results for DNN vs SSGAN. This looses the context of the paper. It would be appropriate to include these three model in the result section then the one presented in the paper. 

Comments on the Quality of English Language

Moderate improvement in the manuscript is required. 

Round 2

Reviewer 2 Report

Comments and Suggestions for Authors

My previous comments were correctly addressed. I suggest the author to put in evidence the contributions in the introduction instead of the conclusions, since the paper's contributions should be evident as soon as possible (referred to the beginning of the reading).

Comments on the Quality of English Language

I suggest a global revision of the english phrasing to avoid any typo.

Author Response

Comment1 : My previous comments were correctly addressed. I suggest the author to put in evidence the contributions in the introduction instead of the conclusions, since the paper's contributions should be evident as soon as possible (referred to the beginning of the reading).

Reply1: Thank you very much for the comment. As the suggestion, a phrase is added in the introduction instead on the conclusion with some new references. You can find the edited part below:

"Machine learning approaches for RSSI-based indoor localization have been reported in studies such as [12–14], and these methods typically depend on manually collected fingerprint-labeled data. As mentioned in [15], collecting fingerprints over a large area is a daunting task, making those methods practically infeasible to apply. Therefore, the accurately produced fingerprint data presented in this paper may significantly support RSSI-based localization. For instance, the synthetic data can be used as an alternative for predicting locations in areas that are otherwise inaccessible. "

12. Zhu, H.; Cheng, L.; Li, X.; Yuan, H. Neural-network-based localization method for Wi-Fi fingerprint indoor localization. Sensors 2023, 23, 6992. 326
13. Tsanousa, A.; Xefteris, V.R.; Meditskos, G.; Vrochidis, S.; Kompatsiaris, I. Combining rssi and accelerometer features for room-level localization. Sensors 2021, 21 , 2723. 
14. Rizk, H.; Elmogy, A; Yamaguchi, H. A robust and accurate indoor localization using learning-based fusion of Wi-Fi RTT and RSSI. Sensors 2022, 22, 2700. 
15. Han, L.; Jiang, L.; Kong, Q.; Wang, J.; Zhang, A; Song, S. Indoor localization within multi-story buildings using MAC and RSSI fingerprint vectors. Sensors 2019, 19, 2433.

Also, I have undertaken a thorough revision of the paper to avoid typos and improve the overall quality of English language.